# Characterization of a Marine Diatom Chitin Synthase Using a Combination of Meta-Omics, Genomics, and Heterologous Expression Approaches

Zhanru Shao,[a,b,c] Osei Ampomah,[d] Fabio Rocha Jimenez Vieira,[b] Richard G. Dorrell,[b] Shaoxuan Li,[e] Leila Tirichine,[b*] Vincent Bulone,[d,f,g] Delin Duan,[a,c] Chris Bowler[b]

aCAS and Shandong Province Key Laboratory of Experimental Marine Biology, Center for Ocean Mega-Science, Institute of Oceanology, Chinese Academy of Sciences, Qingdao, China

bInstitut de Biologie de l'ENS (IBENS), Département de Biologie, École Normale Supérieure, CNRS, INSERM, Université PSL, Paris, France

cLaboratory for Marine Biology and Biotechnology, Qingdao National Laboratory for Marine Science and Technology, Qingdao, China

dDivision of Glycoscience, Department of Chemistry, School of Engineering Sciences in Chemistry, Biotechnology and Health, Royal Institute of Technology (KTH), AlbaNova University Centre, Stockholm, Sweden

eQingdao Academy of Agricultural Sciences, Qingdao, China

fAustralian Research Council Centre of Excellence in Plant Cell Walls, School of Agriculture, Food and Wine, University of Adelaide, Urrbrae, South Australia, Australia

gAdelaide Glycomics, School of Agriculture, Food and Wine, University of Adelaide, Urrbrae, South Australia, Australia

**ABSTRACT** $\beta$-Chitin has important ecological and physiological roles and potential for widespread applications, but the characterization of chitin-related enzymes from $\beta$-chitin producers was rarely reported. Querying against the *Tara* Oceans Gene Atlas, 4,939 chitin-related unique sequences from 12 Pfam accessions were found in Bacillariophyta metatranscriptomes. Putative chitin synthase (CHS) sequences are decreasingly present in Crustacea (39%), Stramenopiles (16%) and Insecta (14%) from the Marine Atlas of *Tara* Oceans Unigenes version 1 Metatranscriptomes (MATOUv1+T) database. A CHS gene from the model diatom *Thalassiosira pseudonana* (Thaps3_J4413, designated Tp*CHS1*) was identified. Homology analysis of Tp*CHS1* in Marine Microbial Eukaryote Transcriptome Sequencing Project (MMETSP), PhycoCosm, and the PLAZA diatom omics data set showed that Mediophyceae and Thalassionemales species were potential new $\beta$-chitin producers besides Thalassiosirales. Tp*CHS1* was overexpressed in *Saccharomyces cerevisiae* and *Phaeodactylum tricornutum*. In transgenic *P. tricornutum* lines, TpCHS1-eGFP localizes to the Golgi apparatus and plasma membrane and predominantly accumulates in the cleavage furrow during cell division. Enhanced Tp*CHS1* expression could induce abnormal cell morphology and reduce growth rates in *P. tricornutum*, which might be ascribed to the inhibition of the G2/M phase. *S. cerevisiae* was proved to be a better system for expressing large amounts of active TpCHS1, which effectively incorporates UDP-*N*-acetylglucosamine in radiometric *in vitro* assays. Our study expands the knowledge on chitin synthase taxonomic distribution in marine eukaryotic microbes, and is the first to collectively characterize an active marine diatom CHS which may play an important role during cell division.

**IMPORTANCE** As the most abundant biopolymer in the oceans, the significance of chitin and its biosynthesis is rarely demonstrated in diatoms, which are the main contributors to the primary productivity of the oceans, ascribed to their huge biomass and efficient photosynthesis. We retrieved genes involved in chitin-based metabolism against the *Tara* Oceans Gene Atlas to expand our knowledge about their diversity and distribution in the marine environment. Potential new producers of chitin were found from the analysis of various algal transcriptome and genome databases. Heterologous expression confirms that *Thalassiosira pseudonana* contains an active chitin synthase (CHS) which may play an important role in the cell division process of diatoms. This study provides new insight

Address correspondence to Delin Duan, dlduan@qdio.ac.cn, or Chris Bowler, cbowler@biologie.ens.fr.

*Present address: Leila Tirichine, Université de Nantes, CNRS, UFIP, UMR 6286, Nantes, France.

The authors declare no conflict of interest.

into CHS geographic and taxonomic distribution in marine eukaryotic microbes, as well as into a new CHS functioning in the biosynthesis of $\beta$-chitin in diatoms.

**KEYWORDS** chitin, chitin synthase, diatom genomes, enzymatic activity, meta-omics, subcellular localization

Chitin is a linear polysaccharide consisting of $\beta$-(1,4)-linked $N$-D-acetylglucosaminyl residues (GlcNAc). It is the second most abundant biopolymer on Earth after cellulose and the most abundant in the marine environment (1). The biosynthesis and degradation of chitin are of significant importance for the biogeochemical cycling of carbon and nitrogen across marine, freshwater, and land ecosystems (2–4). It occurs as two main crystalline allomorphs designated as $\alpha$- and $\beta$-chitin (5). $\beta$-Chitin consists of assemblies of parallel glycan chains, which lead to weaker intermolecular interactions and make $\beta$-chitin more prone to forming deacetyl derivatives, such as chitosan, with enhanced swelling and solubility compared to $\alpha$-chitin (6–8). $\alpha$-Chitin is present in insect cuticles, shells of crustaceans, and fungal cell walls, whereas $\beta$-chitin has been found predominantly in squids and diatoms (9–11).

As $\beta$-chitin natural producers, the diatoms have huge biomass and play an important role in global primary production and are estimated to contribute 20% of primary productivity on Earth (12). With the analysis of recently released genomes and transcriptomes of various diatom species, chitin-based metabolism in diatoms is found to be a complex process (13–16). The Thalassiosirales are so far the only known order of diatoms that produce $\beta$-chitin microfibrils secreted through the fultoportulae (17–19). Chitin fibers extruded from the frustule decrease sinking rates by increasing the resistance of *Thalassiosira* cells in the water column (20, 21). The content of chitin fibers was as high as 31 to 38% of the total cell mass of *Thalassiosira* species (dry weight including the silica) (22). Despite the importance and richness of $\beta$-chitin in Thalassiosirales, the current understanding of the distribution, abundance, and composition of chitin-synthesizing marine eukaryotic microbiomes is scarce. The known meta-omics explorations focused on the degradation and utilization of chitin by marine virus or bacteria (23, 24). The *Tara* Oceans data set has provided the taxonomic diversity of diatoms across the main ocean provinces, with a large number of marine metagenome and metatranscriptome sequences to be discovered (25–27). Whether chitin metabolism exists in other marine diatoms besides Thalassiosirales and its molecular details and corresponding regulatory mechanisms in diatoms wait to be investigated.

Both $\alpha$- and $\beta$-chitins are synthesized by chitin synthase (CHS, EC 2.4.1.16), which converts UDP-$N$-acetylglucosamine into linear chains of $\beta$-(1,4)-linked GlcNAc residues (28). CHS enzymes belong to glycosyl transferase family 2 (GT2) in the Carbohydrate-Active enZYmes (CAZy) database (http://www.cazy.org). CHS genes were first isolated and sequenced from the yeast *Saccharomyces cerevisiae* (29). Genome sequencing has since revealed the broad distribution of CHS genes, and their functions in fungi and insects have been extensively investigated via gene knockout or RNA interference (30, 31). All CHS proteins known so far contain multiple transmembrane domains and are similar to cellulose synthases (32). It is expected that they form a channel in the cell membrane for extruding chitin fibrils into the extracellular space. CHS genes have been reported to mainly evolve via duplications and losses, but CHS catalytic activity has been conserved during evolution (33–35). Additionally, the evolutionary history of eukaryotic CHSs indicates that diatom CHSs have a polyphyletic origin (36). However, almost all CHS enzymes studied to date catalyze the synthesis of $\alpha$-chitin. Durkin et al. first isolated six genes encoding three types of chitin synthases from *Thalassiosira pseudonana* designated class A (Thaps3_J7305 and Thaps3_J7306), class B (Thaps3_J6575), and class C (Thaps3_J4368, Thaps3_J4413, and Thaps3_J4414) (37). Analysis of Thaps3_J4368, Thaps3_J6575, and Thaps3_J7305 revealed that their transcript abundance was correlated with chitin production and cell wall formation (37). Wustmann et al. observed the localization of chs7305 (i.e., Thaps3_J7305) in *T. pseudonana*, and reported that the colocalization of chs7305 with chitin provided evidence for the importance of chitin synthesis for cell wall function (38). However, no further functional characterization of diatom CHSs has been documented.

Considering that Durkin et al. only cloned Thaps_J4413 (designated Tp*CHS1*) but did not perform further molecular experiments to investigate its function (37), here, we

**TABLE 1** Pfam domains potentially related to chitin metabolism in *Tara* Oceans Bacillariophyta metatranscriptomes

| Accession no. | Function | ID | Description | No. of sequences |
|---|---|---|---|---|
| PF01607 | Chitin binding | CBM_14 | Chitin-binding peritrophin-A domain | 1,395 |
| PF00187 | | Chitin_bind_1 | Chitin recognition protein | 269 |
| PF02839 | | CBM_5_12 | Carbohydrate-binding domain | 57 |
| PF00379 | | Chitin_bind_4 | Insect cuticle protein | 8 |
| PF03427 | | CBM_19 | Carbohydrate-binding domain | 1 |
| PF14600 | | CBM_5_12_2 | Cellulose-binding domain | 1 |
| PF03142 | Chitin synthase | Chitin_synth_2 | Chitin synthase | 420 |
| PF01644 | | Chitin_synth_1 | Chitin synthase | 4 |
| PF00704 | Chitinase | Glyco_hydro_18 | Glycosyl hydrolase family 18 | 1,337 |
| PF03067 | | LPMO_10 | Lytic polysaccharide mono-oxygenase, cellulose-degrading | 1,232 |
| PF00182 | | Glyco_hydro_19 | Chitinase class I | 138 |
| PF01522 | Chitin deacetylase | Polysacc_deac_1 | Polysaccharide deacetylase | 77 |

identified and collectively characterized Tp*CHS1* from the $\beta$-chitin producer *T. pseudonana*. The geographic and taxonomic distribution of Tp*CHS1* homologous sequences identified within the *Tara* Oceans data set and three diatom genome resources were analyzed. *Saccharomyces cerevisiae* was used to overexpress TpCHS1 for the validation of its chitin synthase activity via *in vitro* assays. TpCHS1 was also transformed into the model diatom *Phaeodactylum tricornutum*, which contains the same secondary plastids as *T. pseudonana*, to investigate TpCHS1's subcellular localization and its effects on cell growth and division. This is an extensive characterization of an active chitin synthase from diatoms and provides important insights into its effects on nonchitin producers.

## RESULTS

**Retrieval of proteins directly related to chitin in *Tara* Oceans Bacillariophyta metatranscriptomes.** At the EMBL-EBI Pfam website (https://www.ebi.ac.uk/interpro/), we retrieved 14 Pfams for "chitin," among which 4,939 unique sequences from 12 Pfam accessions had been found in *Tara* Oceans Bacillariophyta metatranscriptomes (Table 1). The 12 Pfams contained 4 categories of proteins with a chitinase domain (2,707 distinct sequences), a chitin-binding domain (1,731 distinct sequences), a chitin synthase domain (424 distinct sequences), and a chitin deacetylase domain (77 distinct sequences) (Table 1). For chitin synthases, only 4 hits were obtained for PF01644 (Chitin_synth_1 domain), whereas the other 420 hits were found to contain PF03142 (Chitin_synth_2 domain), which is also present in our TpCHS1 sequence reported here. Figure 1A shows the mRNA levels of genes containing each Pfam domain in samples from different organismal size fractions (0.8 to 5, 5 to 20, 20 to 180, and 180 to 2,000 $\mu$m) across *Tara* Oceans regions. Globally, these Pfam domains were present and expressed in diatoms and covered a broad range of cell sizes represented in each of the size fractions. Notably, in the Mediterranean Sea (MS), PF03142 (Chitin_synth_2 domain) was highly expressed in the 0.8- to 5-$\mu$m size fraction, whereas PF01522 (Polysacc_deac_1) was abundant in the 5- to 20-$\mu$m and 20- to 180- $\mu$m size fractions. Chitinase sequences (PF03067, PF00704, and PF00182) were highly transcribed in larger size fractions, except for those from the Southern Ocean (SO), which had higher expression levels in the 0.8- to 5-$\mu$m size fraction (Fig. 1A; Table S1). Of further note, PF14600 (CBM_5_12_2) was only detected in the Southern Ocean, suggesting a specific role for diatom proteins containing this domain in polar regions.

**Geographic and taxonomic distribution of PF03142 sequences.** The chitin_synth_2 family domain (PF03142) was found in 420 transcripts in the global ocean diatom community. We further assessed the diversity and abundance of these sequences in *Tara* Oceans metagenomes and metatranscriptomes. They were evenly distributed in the northern and southern hemispheres and were almost equally abundant in the different oceans (Fig. 1B). We did not observe ocean areas with unusually high or low diversity/abundance. The chitin_synth_2 family domain was queried against the Ocean Gene Atlas (http://tara-oceans.mio.osupytheas.fr/ocean-gene-atlas/) (39), a web service for exploring the biogeography of genes from marine planktonic organisms, using the Marine Atlas of *Tara* Oceans Unigenes version 1 Metatranscriptomes (MATOUv1+T) database. Analysis of the taxonomic distribution showed that among marine eukaryotic hits, the most abundant group was Crustacea (39%),

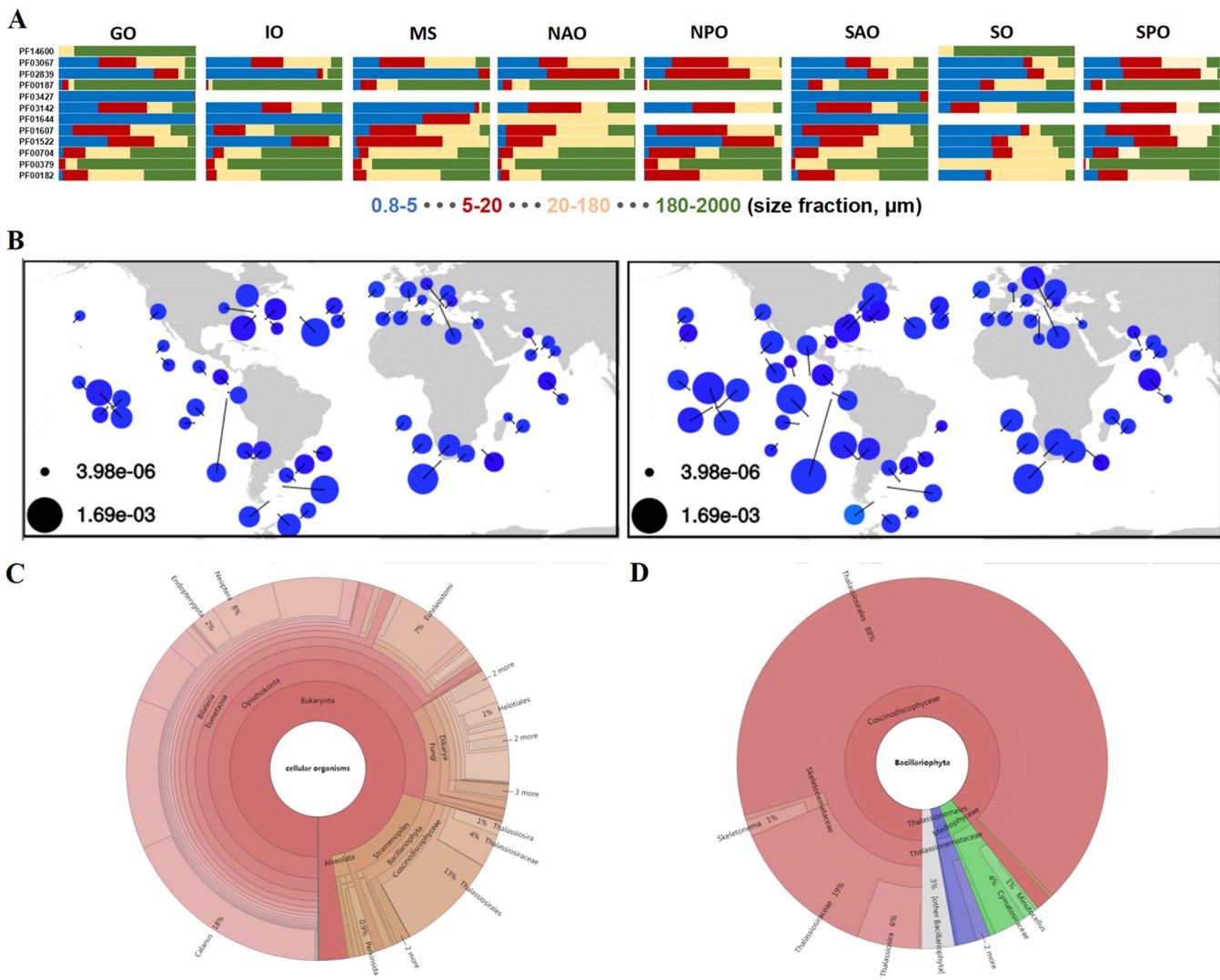

**FIG 1** Pfam data mining of chitin-related genes and chitin_synth_2 domain distribution in *Tara* Oceans metagenome and metatranscriptome data sets. (A) Pfam domains from proteins related to chitin in *Tara* Oceans Bacillariophyta metatranscriptomes across the global ocean. Metatranscriptomic levels were normalized by the total abundance of Bacillariophyta transcriptomes at each station for each of the four size fractions (0.8 to 5, 5 to 20, 20 to 180, and 180 to 2,000 $\mu$m). The details of each Pfam domain are listed in Table 1. (B) Geographic distribution and abundance of the chitin_synth_2 domain (PF03142) in diatoms. The left panel is from the metagenome, and the right panel is from the metatranscriptome. (C) Taxonomic distribution of the chitin_synth_2 domain from the eukaryote metatranscriptome database in the Ocean Gene Atlas website (http://tara-oceans.mio.osupytheas.fr/ocean-gene-atlas/) (39). (D) Taxonomic distribution of the chitin_synth_2 domain in diatoms. GO, Global Ocean; IO, Indian Ocean; MS, Mediterranean Sea; NAO, North Atlantic Ocean; NPO, North Pacific Ocean; SAO, South Atlantic Ocean; SO, Southern Ocean; SPO, South Pacific Ocean.

followed by Stramenopiles (16%), Insecta (14%), Chordata (14%), Fungi (13%), and Alveolata (2%) (Fig. 1C). In Stramenopiles, 89% of the CHS hits were from diatoms (data not shown), and a total of 88% of the diatom hits corresponded to Thalassiosirales, in which the Thalassiosiraceae species accounted for 21% and Skeletonemataceae represented only 2%, and the other 77% were other Thalassiosirales species without further classification (Fig. 1D). In the hits which did not belong to Thalassiosirales, Mediophyceae and Thalassionemales accounted for 5% and 3%, respectively. Within the Mediophyceae, it is worth noting that the polar-centric nanodiatom genus *Minutocellus*, which is distantly related to Thalassiosirales, also contains the chitin_synth_2 domain, indicating that this largely unstudied yet widely distributed group of smaller-sized diatoms (40) may also contain chitin. This result is further supported by the observation that other Pfam domains from diatoms that are potentially involved in chitin metabolism are also enriched in small size fractions, specifically PF02839 (CBM_5_12), PF03427 (CBM_19), and PF01644 (Chitin_synth_1) (Fig. 1A; Table 1).

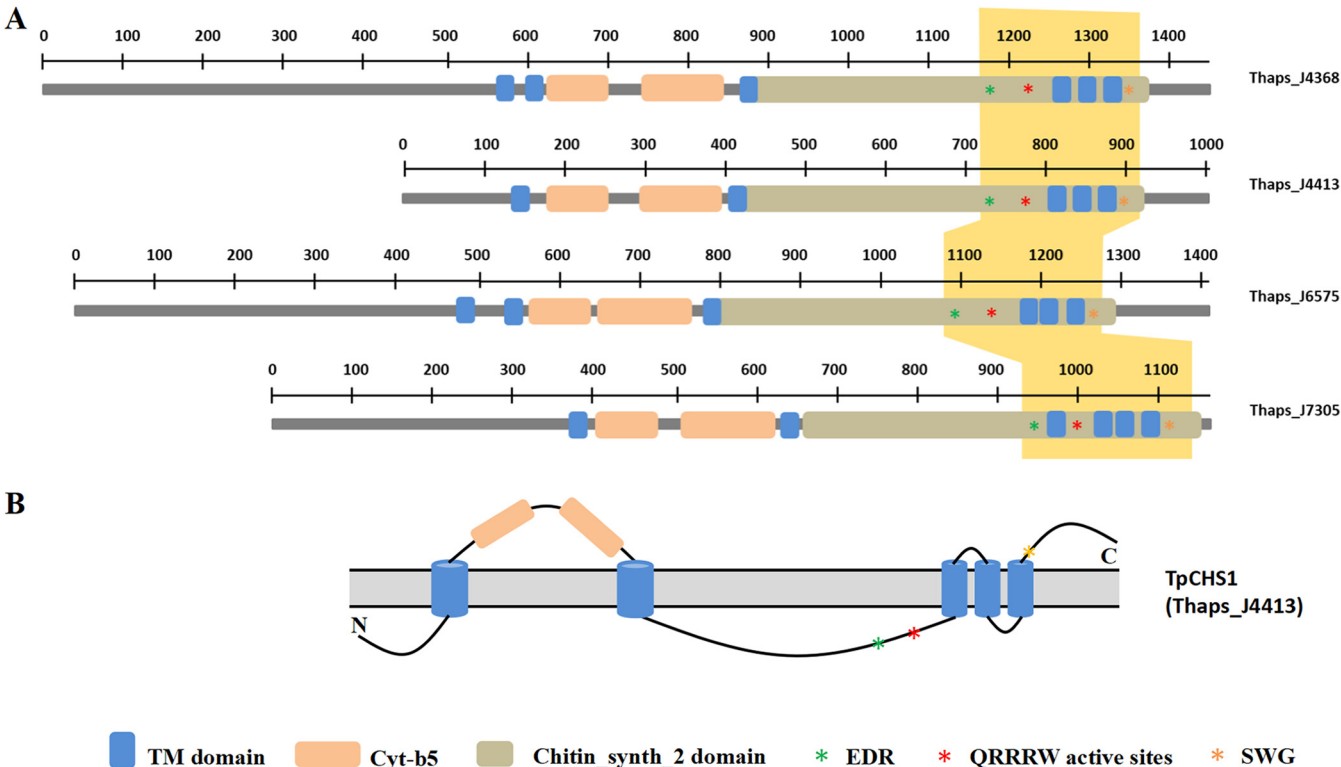

**FIG 2** Schematic representation of the sequence structure of TpCHSs. (A) Comparison of predicted protein domains of cytochrome $b_5$ (cyt-$b5$), transmembrane (TM) helices, and chitin_synth_2 domain among TpCHS sequences. (B) Schematic drawing of the domain organization of TpCHS1. The three conserved motifs, EDR (green asterisk), QRRRW (red asterisk), and SWG (orange asterisk) are located near the C-terminal TM domains.

**Sequence analysis of CHS genes from *T. pseudonana*.** The genome sequences of *T. pseudonana* (i.e., Thaps3) were deposited in the Joint Genome Institute (JGI) database (https://mycocosm.jgi.doe.gov/Thaps3/Thaps3.home.html) (21, 37). However, it still remains unclear how many CHS candidates there are in *T. pseudonana*. Durkin et al. manually annotated six genes encoding putative chitin synthases from the genome of *T. pseudonana*. Here, we reanalyzed the Thaps3 data set and identified 20 genes automatically annotated as "chitin synthase" (IPR004835 in the InterPro database) (see Table S2 in the supplemental material). After removing incorrect annotations, sequences encoding the same proteins, and inverted repeats, four putative chitin synthases (Thaps3_J4368, Thaps3_J4413, Thaps3_J6575, and Thaps3_J7305) were retained (Tables S2 and S3). From the SMART modular analysis results, we found that all these four CHS sequences comprise 5 to 6 predicted transmembrane (TM) helices, two cytochrome $b_5$ (cyt-$b5$) domains (SM01117) and a chitin_synth_2 domain (PF03142) from N to C termini (Fig. 2A). We found three conserved motifs, i.e., EDR, QRRRW, and SWG, in the chitin_synth_2 active domain at the C termini of the four CHS proteins. There are no signal peptides, signal anchors, or plastid/mitochondrial target sequences predicted in any of the four TpCHS sequences (data not shown). We did further analysis on the C-terminal regions (yellow shaded portion in Fig. 2A), which contain the three conserved motifs. Amino acid alignment showed that TpCHS sequences present the highest identities, with CHS3 from *S. cerevisiae* (ScCHS3, NP_009579.1) (Fig. S1A; Table S3). Figure S1B shows a conserved DAD motif in TpCHSs and ScCHS3, which functions in binding metal ions when coordinating the phosphates of the NDP-sugar substrate. Additionally, a ligand binding site of P*Y...D was found in the TpCHS sequences. The results also show that the TpCHS sequences have a closer evolutionary relationship (Fig. S1C) and a similar organization of conserved motifs (Fig. S1D) as ScCHS3 compared to ScCHS1 and ScCHS2. Specifically, a detailed schematic drawing of the domain organization for TpCHS1 was shown in Fig. 2B. The N terminus is predicted to be located in the cytoplasm, and the two cyt-$b5$ and C-terminal domains are on the extracellular side of the membrane. Five TM helices are

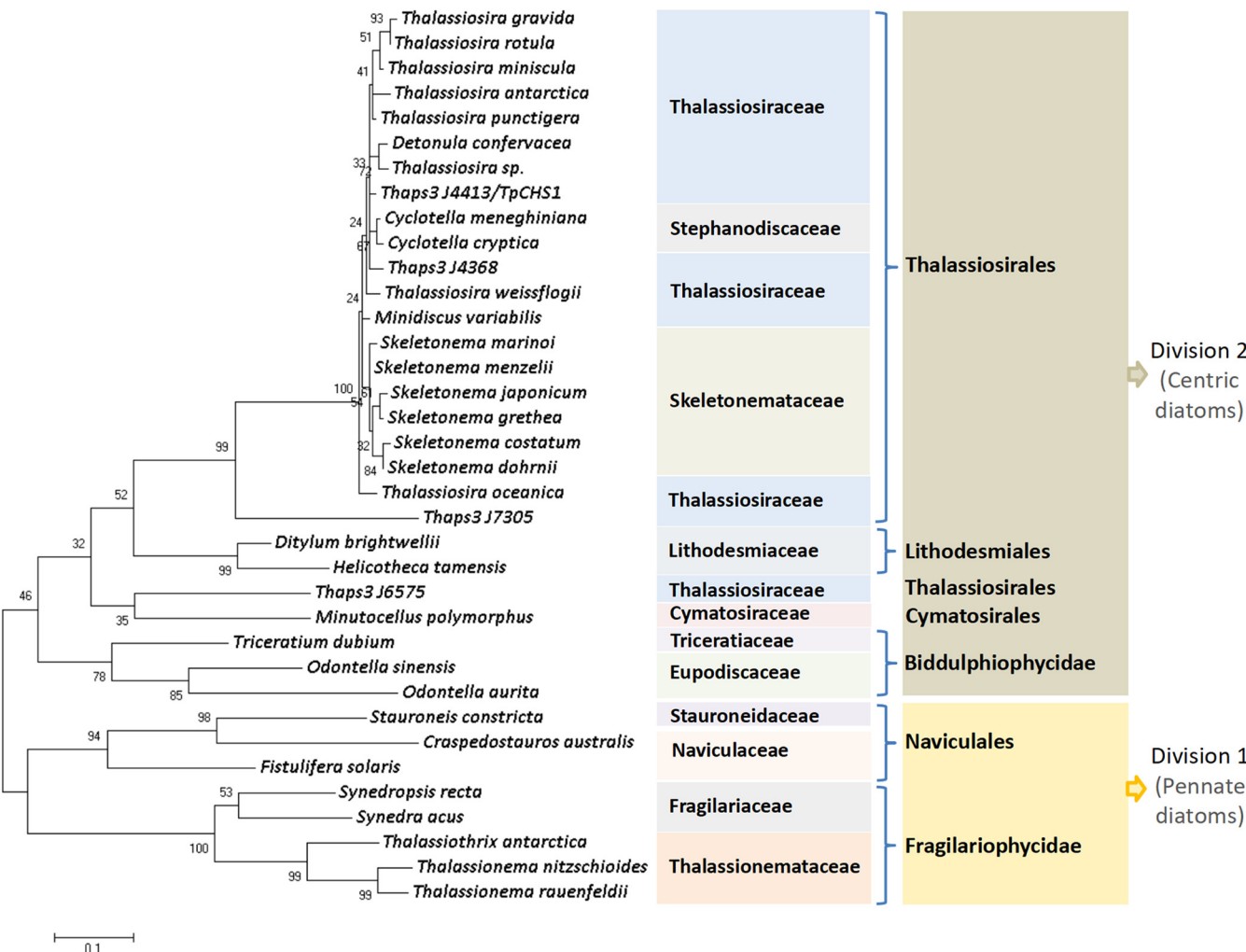

**FIG 3** Phylogenetic tree constructed based on 36 putative chitin synthase amino acid sequences in diatoms. The tree was constructed using the maximum-likelihood algorithm in MEGA 7.0.26 based on the Jones-Taylor-Thornton (JTT) matrix-based model with 1,000 bootstrap replicates. Proteins are from the MMETSP, PhycoCosm, and PLAZA databases, and all 36 sequences are listed in Table S6. All positions containing gaps and missing data were eliminated.

shown as blue cylinders integrated into the plasma membrane. The coding sequence (CDS) of Tp*CHS1* was cloned from *T. pseudonana* cDNA, and after sequencing the PCR product, we verified that the cloned Tp*CHS1* had the same sequence as the genomic Thaps3_J4413 sequence. The sequence structural features of Tp*CHS1* are summarized in Table S4. Tp*CHS1* encodes a protein of 1,003 amino acids (molecular weight [MW], 113.1 kDa; pl, 5.37), which contains a large number of random coils (35.0%) and alpha helices (34.8%).

**Phylogenetic analysis of putative CHSs in the diatom genome and transcriptome databases.** We investigated the evolutionary relationships among diatom species containing CHSs. In this study, 501, 425, and 32 TpCHS1 homologous sequences (putative chitin synthases) were retrieved from the MMETSP, PhycoCosm, and PLAZA data sets, respectively (Table S5). These sequences were highly enriched in Thalassiosirales species (841 hits), consistent with the results from the *Tara* Oceans data (Fig. 1D). In these 841 Thalassiosirales sequences, 697 (82.9%) were from Thalassiosiraceae and 124 (14.7%) were from Skeletonemataceae. More than 10 hits were found from Naviculales (41), Eupodiscales (22), Lithodesmiales (13), and Cymatosirales (10). We arranged these 841 sequences according to sequence identity values with TpCHS1 after BLASTP homologous alignment. Subsequently, a phylogenetic tree was constructed using the 36 sequences with the highest similarity from each homologous species (Fig. 3; Table S6). CHSs are clustered into two individual clades, with 28 sequences from centric diatoms (division 2) and eight sequences from pennate diatoms (division 1) (Fig. 3). TpCHS1

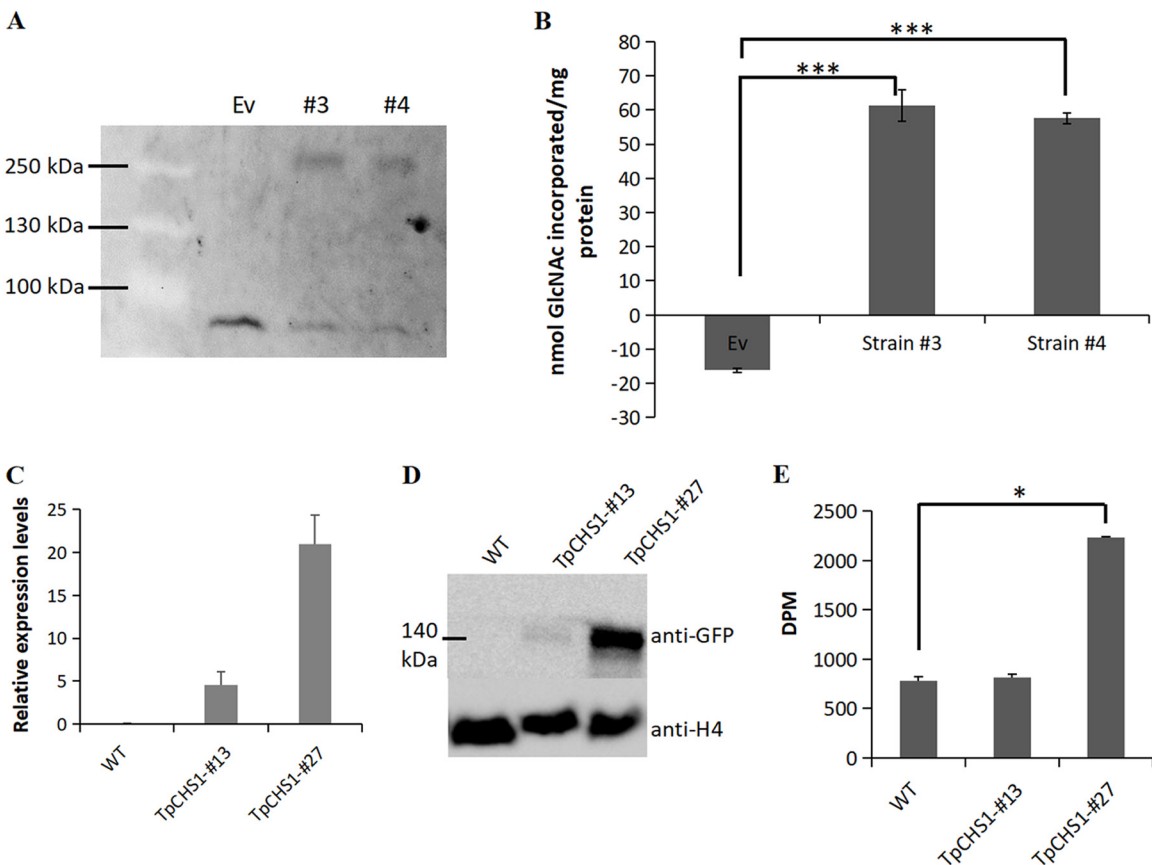

**FIG 4** Overexpression of Tp*CHS1* in *Saccharomyces cerevisiae* BY4742 and *Phaeodactylum tricornutum*. (A) In-gel fluorescence showing the overexpression of TpCHS1-eGFP fusion protein in *Saccharomyces cerevisiae*. The asterisk shows a single nonspecific fluorescent band observed in the samples prepared from the empty vector (Ev) transformant. (B) Chitin synthase activity tested from two yeast positive colonies (no. 3 and no. 4). No activity was detected in insoluble fractions (data not shown). (C) qRT-PCR assay showing high expression of Tp*CHS1* in *Phaeodactylum tricornutum* transformants no. 13 and no. 27. (D) Western blot showing the expression of the TpCHS1-eGFP fusion protein in *Phaeodactylum tricornutum* transformed lines. Proteins were probed with an anti-GFP antibody. Histone H4 was used as a loading control. (E) Chitin synthase activity assay in the two *Phaeodactylum tricornutum* transgenic lines with respect to wild-type cells. The data in panels B, C, and E used one-way ANOVA statistical analysis with Tukey's test. Data are represented as the mean $\pm$ standard deviation (SD) ($n = 3$). **, $P < 0.01$; ***, $P < 0.001$; ****, $P < 0.0001$.

had a close evolutionary relationship with CHSs from *Detonula confervacea*, *Cyclotella meneghiniana*, and *Minidiscus variabilis*, except for *Thalassiosira* CHSs. Members of the genus *Thalassiosira* were distributed across different lineages within the Thalassiosirales. Thaps_J6575 was not grouped into the Thalassiosirales clade but formed an individual cluster with *Minutocellus polymorphus* CHS. Thalassiosirales and Lithodesmiales were sister clades. We found that three symmetric biraphid diatoms, *Stauroneis*, *Craspedostauros*, and *Fistulifera*, and four araphid diatoms, *Synedropsis*, *Synedra*, *Thalassiothrix*, and *Thalassionema*, contained CHS homologous sequences.

**Tp*CHS1* overexpressed in *S. cerevisiae* exhibits chitin synthase activity *in vitro*.** Although TpCHS1 contains the chitin_synth_2 domain, the ability of GlcNAc incorporation is the most direct evidence that it is indeed a chitin synthase. A radiometric *in vitro* activity assay was performed to authenticate the chitin synthase activity of TpCHS1. Two positive *S. cerevisiae* lines (no. 3 and no. 4) expressing Tp*CHS1* were retained, and their microsomal fractions were harvested by differential centrifugation. The in-gel fluorescence assay (Fig. 4A) shows a clear band with an estimated molecular weight of 140 kDa for both transgenic lines and a single nonspecific band (70 kDa) from the empty vector (Ev) transformant, which verifies the successful overexpression of the Tp*CHS1* gene. A radiometric *in vitro* activity assay using $^{14}$C-labeled UDP-*N*-acetyl-$_D$-glucosamine and digitonin-solubilized microsomal fractions indicated that the recombinant enzyme displays high chitin synthase activity compared to the empty vector control (Fig. 4B).

**Tp*CHS1-eGFP* expression in *P. tricornutum* induces phenotypic changes.** *In situ* observation of TpCHS1 protein is of great importance for its functional investigation. In this study, we verified that it was very challenging to do biolistic transformation on *Thalassiosira* cells. Considering the similar plastid evolution (secondary endosymbiosis) and structure (bounded by four membranes) between *Phaeodactylum* and *Thalassiosira*, together with the fact of *P. tricornutum* being a well-established lab model, we introduced a fusion of Tp*CHS1* with a C-terminal enhanced green fluorescent protein (eGFP) gene (Tp*CHS1-eGFP*) into *P. tricornutum* by gene bombardment. Quantitative real-times PCR (qRT-PCR) assays showed that the Tp*CHS1* genes from two transgenic lines (no. 13 and no. 27) were highly transcribed, whereas transcripts could not be detected in wild-type cells (Fig. 4C). Immunoblots with an anti-GFP antibody revealed a strong band of the expected size (~140 kDa) in the Tp*CHS1-eGFP* transgenic line no. 13 and a stronger band in no. 27 (Fig. 4D). The activity assay against $^{14}$C-labeled UDP-*N*-acetyl-ᴅ-glucosamine using digitonin-solubilized microsomal fractions indicated that the recombinant enzyme from Tp*CHS1-eGFP* no. 27 displays high chitin synthase activity, with a disintegration per minute (DPM) value much higher than that detected in wild-type (2.9-fold) and Tp*CHS1-eGFP* no. 13 (2.7-fold) cells (Fig. 4E). Although *P. tricornutum* was capable of expressing Tp*CHS1* with measurable chitin synthase activity, its activity was only 1-20th of that measured in our *S. cerevisiae* samples.

Morphological phenotypes were observed between wild-type and transgenic lines no. 13 and no. 27. Most cells from line no. 13 did not exhibit an abnormal phenotype like no. 27 (data not shown). We presume that the amount of TpCHS1-eGFP fusion protein expressed in no. 13 was not sufficient to cause changes in cell morphology. For the Tp*CHS1-eGFP* #27 cell line, the cells were shorter than their wild-type counterpart and *pPha-T1-eGFP* (empty vector) control cells, and both cellular ends were more obtuse (Fig. 5A). We found a particularly high number of abnormal cells when Tp*CHS1-eGFP* no. 27 cells were undergoing cell division (Fig. S2A). Figure S2B shows the localization of TpCHS1 in abnormal cells. To assess whether the presence of the TpCHS1-eGFP fusion protein affected cell growth rates, we determined their growth characteristics over a prolonged growth cycle. Figure 5B shows that the Tp*CHS1-eGFP* transgenic cells grew slower than wild-type and empty vector controls.

We measured the transcriptional levels of three cell cycle marker genes, *CYCP6*, *E2F1*, and *CYCB1*, to determine which phase was retarded due to the overexpression of Tp*CHS1*. Figure 5C shows that *CYCP6* and *E2F1* are stably transcribed in the G1 (darkness) and S (3 h of illumination) phases, respectively. In contrast, *CYCB1* is highly expressed in the G2/M phase (6 to 9 h of illumination) in the transgenic cells, with an increase of 1.4-fold compared to the wild type (Fig. 5C). This result indicates that Tp*CHS1-eGFP* transgenic cells grow more slowly, primarily due to changes in the G2/M cell cycle phase.

**Subcellular localization of TpCHS1 is highly correlated with the cell cycle.** We further assessed the subcellular localization of TpCHS1 in *P. tricornutum* transgenic lines no. 13 and no. 27. The location of the TpCHS1-eGFP fusion protein was highly correlated with cell cycle progression (Fig. 6). Before cells start dividing, GFP fluorescence accumulates around the cell and next to the chloroplast (Fig. 6A). Subsequently, the GFP signal moves toward both tips of the daughter chloroplasts and relocalizes to the cleavage furrow (Fig. 6B). With cell division, the signals are observed near the chloroplasts in the middle of the daughter cells (Fig. 6C). Additionally, the TpCHS1 signal (green) and the nuclei signal (blue, DAPI staining) cooccur near the chloroplast, but the signals do not overlap (Fig. 6D). Although the no. 13 cell line did not express a high level of TpCHS1-eGFP, the TpCHS1 signal also appeared near the plastid but not in the nucleus (Fig. S2C). According to the various compartments studied in *P. tricornutum* by Liu et al. (41), we propose that TpCHS1 might undergo complex protein processing in the Golgi apparatus before becoming a mature transmembrane protein.

## DISCUSSION

**New insights into the taxonomic distribution of CHS in diatoms.** Chitin has been widely found in different ecosystems in aqueous and terrestrial environments. Microbial meta-omics encompass the sequence repository of various communities, e.g., Terrestrial Metagenome DB, including 15,022 terrestrial metagenomes which do not belong to marine environments (42). Considering the scarce cases for the exploration of chitin-relevant genes

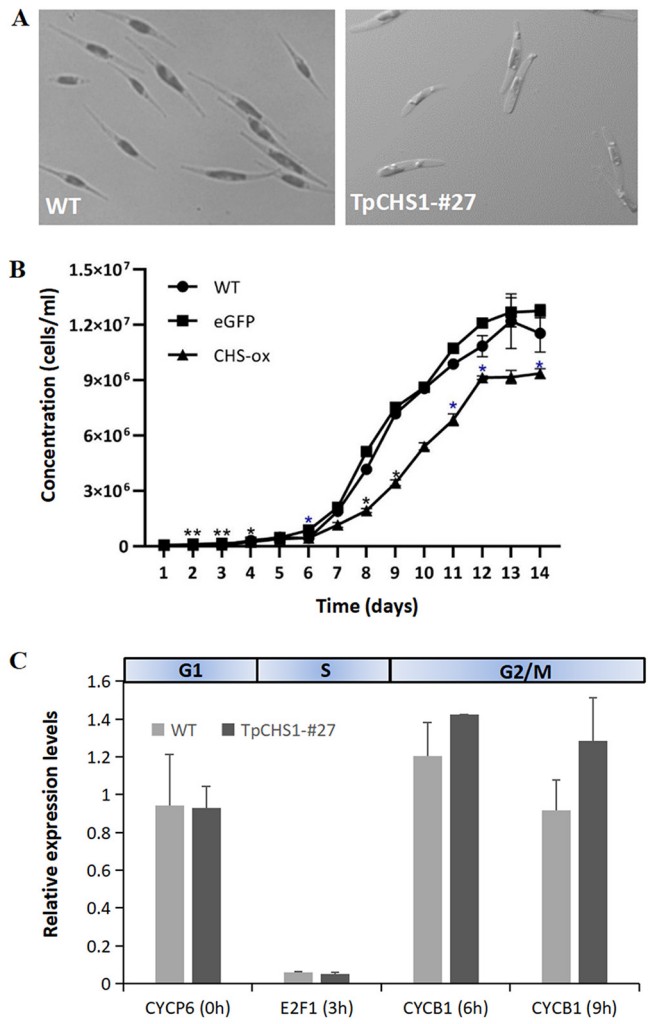

**FIG 5** Tp*CHS1-eGFP* expression in *P. tricornutum* caused abnormal cell morphologies and constrained cell growth. (A) Phenotypes of the wild-type and the Tp*CHS1-eGFP* transgenic line no. 27 showing shorter cells with obtuse tips. Scale bars = 5 $\mu$m. (B) Growth curves of WT, p*PhaT1-eGFP*, and the Tp*CHS1-eGFP* transgenic line no. 27. *, $P < 0.05$; **, $P < 0.01$. Black asterisk, WT versus CHS-ox; blue asterisk, eGFP versus CHS-ox. (C) Transcriptional profiles of cell cycle marker genes (*CYCP6*, *E2F1*, and *CYCB1*) in Tp*CHS1-eGFP* transgenic cells and wild-type cells during *Phaeodactylum tricornutum* cell cycle progression.

in the ocean, we used the *Tara* Oceans data set, a global ocean atlas of eukaryotic genes, to retrieve chitin synthase (CHS). The *Tara* Oceans data set has revealed that approximately two-fifths of putative CHS genes are from crustaceans in the ocean. However, it is worth noting that Stramenopiles come in second, containing more CHS hits than fungi, although the latter has been considered to be the major source of CHSs (62%) in the RefSeq database. Additionally, our study has provided new information regarding CHS distribution in diatoms, which account for 89% of species of Stramenopiles. *Tara* Oceans metatranscriptomes and CHSs from the MMETSP, PhycoCosm, and PLAZA databases gave consistent results, reflecting that the most abundant CHS homologous sequences in diatoms were from Thalassiosirales. The CHS sequence data set was principally occupied by sequences from marine-centric diatoms, and this is consistent with the result of CHS gene retrieval against the NCBI database by Novis et al. (43). Durkin et al. summarized the known chitin producers, which were predominantly concentrated in *Thalassiosira* (37). Interestingly, a large number of CHSs (242 hits) were found in the genome of the smallest known centric diatom *Minidiscus variabilis*, indicating its potential capability of synthesizing chitin. This was in accordance with the scanning electron microscopy result showing that *Mytilus chilensis* formed a "thread" (presumably chitin) extruded from a central fultoportula (44).

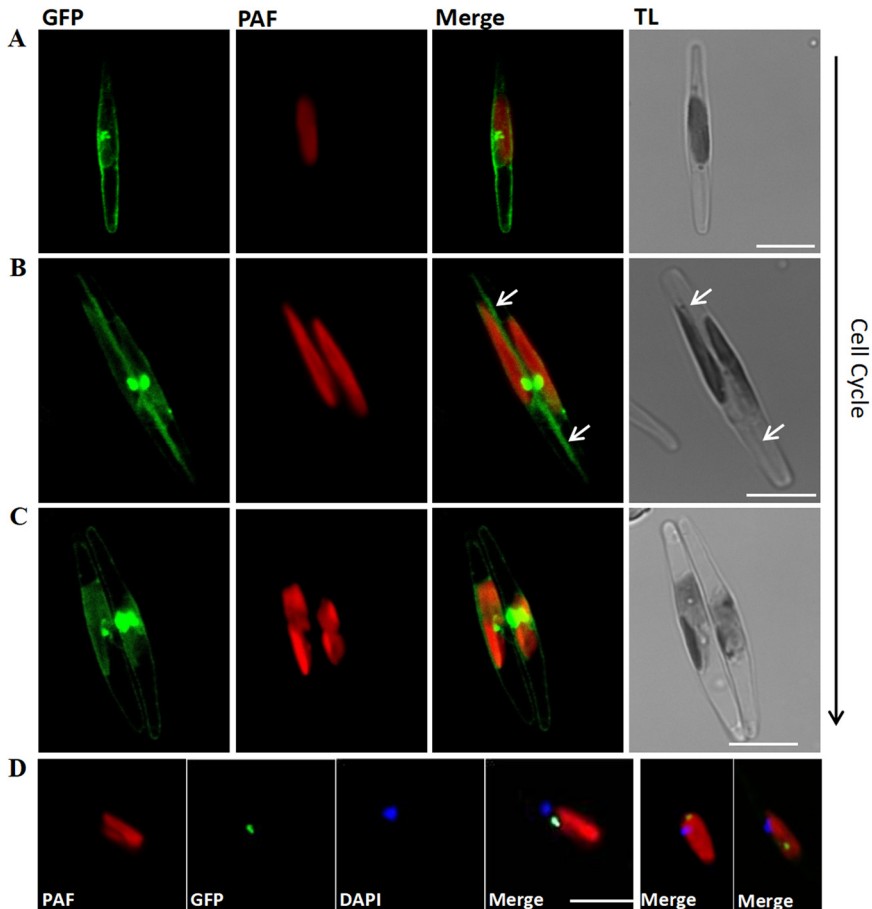

**FIG 6** Subcellular localization of TpCHS1 in *Phaeodactylum tricornutum* cells during cell cycle progression. (A) Cell with the focus on a nondividing chloroplast, displaying the GFP signal (green) near the middle of the chloroplast and around the cell. (B) Cell with the chloroplast completing division, showing the GFP signal moving toward the ends of the chloroplast and accumulating at the cleavage furrow (white arrows). (C) Separating daughter cells showing the GFP signal predominantly next to the chloroplasts. (D) DAPI (4′,6-diamidino-2-phenylindole) staining (blue) showing that the position of nuclei in Tp*CHS1-eGFP* transgenic cells does not overlap TpCHS1 localization (green). All images are from Tp*CHS1-eGFP* transgenic line no. 27, and the location of TpCHS1 in no. 13 is shown in Fig. S2C. GFP (green), green fluorescent protein; PAF (red), plastid autofluorescence; TL, transmission light. Scale bars = 5 $\mu$m.

In our study, we also retrieved CHS genes in the transcriptomes of other potential chitin producers from the polar centric nanodiatom *Minutocellus*, as well as in chain-forming pennate diatoms, such as *Thalassiothrix* and *Thalassionema*. We found four putative CHS genes in *Cyclotella meneghiniana*, which has been reported to contain abundant chitin-related genes and to produce extracellular $\beta$-chitin nanofibers (13, 45). The evolutionary position of *C. meneghiniana* is adjacent to that of *T. pseudonana*. This was in accordance with the results of Alverson et al., who reported that *T. pseudonana* likely descended from a freshwater ancestor in *Cyclotella* (46). Additionally, Sato et al. studied the evolutionary relationships of 106 taxa based on 35 benchmarking universal single-copy orthologs (BUSCOs) in diatoms, and also found that *T. pseudonana* and *C. meneghiniana* are sister taxa (47). Our analysis has also consistently found Thalassiosirales as a sister clade to Lithodesmiales species (17). Previously, an independent origin between *T. pseudonana* and *P. tricornutum* CHS genes was proposed by Gonçalves et al. and Morozov and Likhoshway (35, 36). Here, we expanded CHSs to all available diatom sequences that have been retrieved thus far and reached the consistent conclusion that CHSs from centric diatoms (division 2) and pennate diatoms (division 1) are phylogenetically distinct and may have different ancestral origins. In pennate diatoms, we first reported the presence of CHS homologous sequences in *Stauroneis*, *Craspedostauros*, *Fistulifera*, *Synedropsis*, *Synedra*, *Thalassiothrix*, and *Thalassionema*, which

expanded the taxonomic distribution of CHS in diatoms and provided new sources of novel CHS enzymes. It is worth mentioning that although we found more diatom CHSs classified into multiple divisions, the core or accessory function of these CHS enzymes needs further investigation.

**Sequence structure and enzymatic activity comparisons among various CHSs.** The predicted domain structures of TpCHSs are consistent with the observation made by Fernandes et al. that all CHSs are transmembrane proteins with the catalytic domain located on the cytoplasmic side (34). TpCHSs are presumed to be members of the class IV chitin synthases due to the absence of a chitin_synth_1 domain and their greater sequence identities, closer evolutionary relationship, and similar motif organization with ScCHS3 compared to ScCHS1 or ScCHS2 (48). ScCHS3 contributes in excess of 90% of the total cellular chitin in *S. cerevisiae* and is responsible for the deposition of chitin all over the cell wall, whereas ScCHS1 is involved in repair functions at the end of cytokinesis and ScCHS2 functions in the primary septum formation (49–51). It awaits further experiments to verify if TpCHS1 in diatoms exerts a similar function as ScCHS3 in fungi, due to their similar domain structure and closer evolutionary position.

As chitin is an important composition of the fungal cell wall, CHS has been considered an attractive and green target for fungicides (52). Thus, many studies have been focused on designing, synthesizing, and screening CHS inhibitors (53, 54). However, very few experiments have been conducted to explore novel CHSs to catalyze the formation of chitin. There are so far no studies that have demonstrated CHS activity in algae. Generally, such experiments are proven to be very challenging because of the presence of multiple transmembrane helices in CHSs. Our radiometric *in vitro* activity assay first showed chitin synthase activity in both Tp*CHS1* heterologous expression systems, i.e., the *S. cerevisiae* mutant (*chs1Δ* and *chs3Δ*) and *P. tricornutum*. Excluding the influence of the background activity in wild-type cells, the *in vitro* activity of TpCHS1 expressed in *S. cerevisiae* (our study) was approximately six times higher than that of *Saprolegnia monoica* CHS (SmCHS2) expressed in *Pichia pastoris* (55). This implies that diatom-derived CHSs might be more efficient than those from oomycetes.

**TpCHS1 has the potential for synthesizing chitin in the cell wall of *T. pseudonana*.** In the description of PF03142 (Chitin_synth_2), the inactive form of CHS (zymogen) is first packaged into chitosomes in the Golgi apparatus and then transferred to the interior side of the yeast cell plasma membrane prior to activation. This process is assumed to be the same in centric diatoms since our time course subcellular localization observations showed that TpCHS1 appeared in the Golgi apparatus before locating at the cell wall interface with the intracellular space. The GFP signal is in accordance with the study of the location of ScCHS3, which transports from the trans-Golgi network (TGN) to the plasma membrane (56). This is an indication that TpCHS1 might have a similar function as ScCHS3.

The cell cycle-related location of TpCHS1 in this work is similar to the results of our previous study of diatom chitin deacetylases (PtCDA and TpCDA) (14). In both cases, the proteins did not reside in a specific organelle but were transported within the cytoplasm and accumulated in the cell division plane during cell cycle progression. As an important structural component of the cell wall, chitin production was abundant during cell division, and the cell division plane was observed to be composed of polysaccharide-based fibrillar material (37, 57). We thus propose that diatom CHSs and CDAs might function together where chitin is speculated to occur, when the nascent cell wall forms. This hypothesis is reminiscent of a Thaps3_J7305 (chs7305) study, where chs7305 was observed during the formation of a chitin-based substance at the girdle band region of elongated cells (38). We also observed chitin localization in *T. pseudonana* and found that chitin existed in the cell wall and was highly enriched when cells aggregate (Fig. S2D). This indicates that TpCHS1 might function near the cell wall to produce chitin, and the accumulation of chitin could induce cell aggregation. However, the fluorescence of TpCHS1 (this study), chs7305 (38), and TpCDA (14) did not appear in the elongation of chitin fibers. It will be of interest to further explore the function of CHSs and CDAs by knockdown or knockout experiments.

*S. cerevisiae* was proved in our study to be an effective system for expressing large amounts of active TpCHS1, which is a better model for CHS enzymatic activity detection than overexpression in *P. tricornutum*. However, diatom cells could provide credible

information on cell phenotype variation and potential locations of protein. Choosing *S. cerevisiae* and *P. tricornutum* for heterologous transformation in this study has its own strengths. It is worth mentioning that this work is flawed because of the lack of a second strong transgenic *P. tricornutum* cell line no. 27, and gene editing experiments will be necessary for further elucidation of TpCHS1 function.

In summary, we have identified a chitin synthase gene from a centric diatom and heterologously overexpressed it in *S. cerevisiae* and *P. tricornutum* for functional characterization. Our analysis of the sequence distribution and expression, sequence structure, evolutionary position, protein localization, and enzymatic activity of TpCHS1 represents an extensive investigation of an active chitin synthase from diatoms. Enhanced Tp*CHS1* expression induced abnormal cell morphology and reduced growth rates in *P. tricornutum*, which needs further functional investigation to explore the underlying regulation mechanism.

## MATERIALS AND METHODS

**Diatom culture conditions.** *P. tricornutum* Bohlin (Pt1 8.6; CCMP 2561) and *T. pseudonana* (CCMP 1335) cells were grown in f/2 and enhanced seawater, artificial water, (ESAW) liquid media (58, 59), respectively, at 19℃ in 12-h light/12-h dark diurnal cycles (100 $\mu$mol m$^{-2}$ s$^{-1}$).

**Pfam domain analysis in *Tara* Oceans Bacillariophyta metagenomes and metatranscriptomes.** Detection of chitin-related proteins was performed with the HMMER algorithm using the reference database Pfam 32.0 (http://pfam.xfam.org/, accessed on 20 November 2019) to search the diatom gene atlas generated by *Tara* Oceans (27). Only *Tara* Oceans sequences matching the chitin Pfam families (PF00187, PF02839, PF14600, PF03067, PF18416, PF08329, PF00182, PF01522, PF00379, PF01644, PF00704, PF03142, PF01607, and PF03427) above the default HMMER statistical significance cutoff were considered positive.

**Cloning and sequence analysis of Tp*CHS1* from *T. pseudonana*.** Total RNA extraction and cDNA synthesis from *T. pseudonana* cells at the exponential phase were prepared according to the method of Shao et al. (14). According to the CHS gene sequences described in the JGI genome browser for *T. pseudonana* and the cloned CHS sequences described by Durkin et al. (37), we examined the *in silico* predicted open reading frame (ORF) of the putative CHS gene (Tp*CHS1*, Thaps3_J4413) by cDNA sequencing. The following primers were designed: Tp*CHS1*-Fw, 5′-ATGGACGAAACCTACGCCAG-3′; Tp*CHS1*-Rv, 5′-AAAGTTGATTCTAGACTGCGG-3′. High-fidelity PCR products were amplified using Phusion DNA polymerase (Thermo, USA), and each product was verified by Sanger sequencing (GATC Biotech, France).

To assess the diversity and abundance of chitin synthase sequences across the global ocean, the chitin_synth_2 family domain was queried against the Ocean Gene Atlas website (http://tara-oceans.mio.osupytheas.fr/ocean-gene-atlas/, accessed 30 November 2022) (39), using the Marine Atlas of *Tara* Oceans Unigenes version 1 Metatranscriptomes (MATOUv1+T) database with an E value threshold of 1E$^{-10}$. For the phylogenetic analysis, diatom sequences highly homologous to the full-length amino acid sequence of TpCHS1 were retrieved from the three available genome resources of the MMETSP (60) (data set downloaded and reorganized by coauthor R.G.D., accessed 1 December 2022), PhycoCosm (61) (https://phycocosm.jgi.doe.gov/pages/blast-query.jsf?db=phycocosm, accessed 1 December 2022), and PLAZA (https://bioinformatics.psb.ugent.be/plaza/versions/plaza_diatoms_01/blast/index, accessed 1 December 2022) databases. Homology comparison of TpCHSs and ScCHSs was conducted using the Clustal Omega program with data available from the NCBI website (https://www.ebi.ac.uk/Tools/msa/clustalo/, accessed 1 December 2022). MEGA 7.0.26 was used to construct the maximum-likelihood phylogenetic tree based on the Jones-Taylor-Thornton (JTT) matrix-based model with 1,000 bootstrap replicates (62). Domain topology of TpCHS1 was predicted with the online ProDom and SMART software (http://smart.embl-heidelberg.de, accessed 30 November 2022) (63, 64). The motif elicitation was done using MEME Suite 4.12.0 (https://meme-suite.org/meme/tools/meme, accessed 17 July 2021), followed by the redrawing of the motif pattern with TBtools_JRE1.6 (https://github.com/CJ-Chen/TBtools/blob/master/TBtools_JRE1.6.jar, accessed 17 July 2021). The presence of transmembrane helices was predicted using the online software TMHMM v.2.0 (http://www.cbs.dtu.dk/services/TMHMM/, accessed 25 August 2021) and Phobius (https://www.ebi.ac.uk/Tools/pfa/phobius/, accessed on 25 August 2021).

**Tp*CHS1*-pDDGFP-2 fusion plasmid construction and its overexpression in *S. cerevisiae*.** Tp*CHS1* gene synthesis was conducted using GeneArt with codons optimized for yeast (Thermo Fisher, USA). One pair of primers was designed: Tp*CHS1*-HR-Fw, accccggattctagaactagtggatccccATGGACGAAACCTACGCTTC; Tp*CHS1*-HR-Rv, aaattgaccttgaaaatataaattttccccGAAGTTGATTCTAGATTCAG. The amplified products and linearized pDDGFP-2 vector (with URA selection marker) were then transformed into *S. cerevisiae* BY4742 (*chs1*Δ::*KanMX4*, *chs3*Δ::*HIS3*, and *pep4*Δ::*LYS2*) competent cells. Table S7 lists the detailed procedures for Tp*CHS1*-pDDGFP-2 plasmid construction. The protocols for preparation of competent cells and transformation are described in Drew et al. (65). The cells were spread onto a –URA selective plate, and positive colonies were inoculated in 10 mL –URA medium with 2% glucose, at 280 rpm and 30℃. The overnight culture was diluted to an optical density at 600 nm (OD$_{600}$) of 0.10 to 0.12 in –URA medium with 0.1% glucose. The expression of the TpCHS1-pDDGFP-2 fusion protein was induced by adding 20% galactose (final concentration of 2%) after the OD$_{600}$ of the culture reached 0.6. The cells were centrifuged at 3,000 $\times$ *g* for 5 min after induction for 22 h, and the resulting pellets were subjected to in-gel fluorescence detection and radiometric assays.

**In-gel fluorescence from yeast cultures expressing Tp*CHS1*-pDDGFP-2.** The pellet was resuspended in 500 $\mu$L yeast suspension buffer (YSB) containing protease inhibitor cocktail. Cells were disrupted with a TissueLyser II instrument (Qiagen, Germany) and centrifuged at 12,000 rpm. The pellet was resuspended in YSB

and reextracted as described above. The supernatant was centrifuged at 22,000 × *g* for 70 min at 4°C. Then, 15 μL of YSB was used to resuspend the new pellet. After being mixed with an equal amount of SB, the mixture was incubated at 37°C for 5 min and loaded on a 10% Tris Glycine gel (Invitrogen, Waltham, MA, USA) for SDS-PAGE analysis. The gel was visualized using an LAS-1000 luminescent image analyzer (Fujifilm, Tokyo, Japan).

**Radiometric *in vitro* activity assay using microsomal fractions.** Cultured *S. cerevisiae* cells (1 L) were processed to obtain microsomal fractions using the method previously described (55). The pellets containing the microsomal fractions obtained after ultracentrifugation at 160,000 × *g* for 2 h were resuspended in buffer composed of 10 mM Tris-HCl buffer, pH 7.0, containing 10% (vol/vol) glycerol and protease inhibitor (Roche, USA). The protein was then adjusted to a final concentration of 3 mg/mL using the same buffer and solubilized in the presence of a final concentration of 0.5% digitonin (Merck, Germany) for 30 min with gentle stirring. Following solubilization, the samples were subjected to an ultracentrifugation for 30 min, and the supernatant was collected. The protein concentration in this fraction was measured using the Bradford assay (66). Microsomal fractions from empty vector cells were prepared in parallel. *P. tricornutum* cell pellets were resuspended in CRB buffer (50 mM Tris-HCl, pH 7.6, 1 mM EDTA, 0.6 M sorbitol, and a protease inhibitor cocktail). The microsomal fraction preparation was prepared as described for the yeast cells.

The *in vitro* chitin synthase activity assays were performed in a total volume of 200 μL, containing 10 mM Tris-HCl, pH 7, 5 mM $MnCl_2$, 0.5 mM UDP-*N*-acetyl-D-glucosamine, 0.04 μCi $^{14}$C UDP-*N*-acetyl-D-glucosamine (300 mCi/mmol), 20 mM *N*-acetylglucosamine, 1 μg/mL trypsin, and 200 μg of solubilized protein. The reaction was allowed to proceed overnight and was stopped with 400 μL of 95% ethanol to detect insoluble chitin products or with chloroform-water (300:100 μL) to detect soluble products, using the procedures described by Brown et al. (67). Product formation was analyzed with a MicroBeta2 scintillation counter (Perkin Elmer, USA) using 3.5 mL Ultima Gold F scintillation liquid for insoluble products and 3 mL Ultima Gold scintillation liquid for soluble products (67).

**Plasmid construction, biolistic transformation, and Western blots.** The Gibson assembly strategy was applied to construct the Tp*CHS1-eGFP* fusion plasmid (NEB, USA) (68). The Tp*CHS1* full-length sequence was amplified from the *T. pseudonana* cDNA products with the following primers: Tp*CHS1*-fcpA-Fw, 5′-caaatttgtctgccgtttcgagaaATGGACGAAACCTACGCCAG-3′; Tp*CHS1*-eGFP-Rv, 5′-cctcgcccttgctcaccatAAAGTT GATTCTAGACTCGG-3′. High-fidelity PCR products were amplified using Phusion DNA polymerase (Thermo, USA). Tp*CHS1* was inserted into the linearized pPhaT1-eGFP vector fragments as described in the detailed protocol (Table S7). The Tp*CHS1-eGFP* fusion plasmid was integrated into the genome of *P. tricornutum* by gene bombardment. The bombarded cells were plated onto f/2 solid plates (1% Bacto agar) with 100 μg/mL phleomycin. Positive colonies were inoculated into liquid f/2 medium with 100 μg/mL zeocin. The expression of Tp*CHS1* was analyzed on 10% SDS-PAGE gels, and Western blotting was performed using a mouse anti-eGFP primary antibody and a horseradish peroxidase (HRP)-conjugated anti-mouse secondary antibody immunoglobulin G as previously described in Shao et al. (14).

**Phenotypic observations.** Chitin-binding protein (CBP) tagged with eGFP (CBP-eGFP) was used to visualize the localization of chitin in *T. pseudonana* (69). A TCS SP8 confocal microscope (Leica, Germany) with an excitation wavelength of 488 nm and emission wavelength of 510 to 540 nm was used to observe the GFP signal. After cell synchronization in prolonged darkness (70), the *P. tricornutum* wild-type, empty *pPhaT1-eGFP* vector control and Tp*CHS1-eGFP* transgenic lines were cultivated for 14 days, and cell densities were measured using the same protocol of Shao et al. (14). Two replicates were performed for each cell line.

**Quantitative real-time PCR (qRT-PCR).** To investigate the transcriptional profiles of cell cycle marker genes, *P. tricornutum* cells at the exponential growth phase were synchronized in extra 12-h darkness and exposed to light. Cell samples were collected for analysis at 0 h, 3 h, 6 h, and 9 h after light exposure. cDNA templates were prepared according to the method of Shao et al. (14) and adjusted to the same concentration. qRT-PCR was performed with the SYBR Premix *Ex Taq* II kit (TaKaRa, Japan) on the TP800 thermal cycler Dice (TaKaRa, Japan). A reaction volume of 25 μL contained 12.5 μL premix, 2 μL cDNA, 1 μL of each primer (10 μM), and 8.5 μL double-distilled water (ddH₂O). The two-step thermal cycling protocol was 95°C for 30 s, followed by 40 cycles of 95°C for 5 s and 59°C for 30 s. The ribosomal protein small subunit 30S (RPS) gene was used as a reference gene (71). The qRT-PCR primer sequences used in this study are listed in Table S8. The specificity of primers was determined by relevant dissociation curve. Triplicate repeats were performed, and the relative quantitative values were calculated with the $2^{-\Delta\Delta CT}$ method.

**Statistical analysis.** One-way analysis of variance (ANOVA) or two-way ANOVA statistical analysis between multiple groups was performed depending on whether the experiment contained one factor or two factors, followed by Tukey's or Holm-Sidak's multiple-comparison tests, respectively.

**Data availability.** The data that supports the findings of this study are available in public omics databases and in the supplementary materials of this article. The detailed parameters of the bioinformatics analysis for both online tools and local software have been uploaded to a Zenodo archive (https://zenodo.org/record/7388969#.Y42vkKhByUl; readme.txt). The plasmids applied in this work are available for research use with the permission of coauthors V.B. (pDDGFP-2 vector) and C.B. (pPhaT1-eGFP vector).

## SUPPLEMENTAL MATERIAL

Supplemental material is available online only.
**FIG S1**, PDF file, 0.1 MB.
**FIG S2**, PDF file, 0.3 MB.
**TABLE S1**, PDF file, 0.2 MB.
**TABLE S2**, PDF file, 0.1 MB.
**TABLE S3**, PDF file, 0.1 MB.

**TABLE S4**, PDF file, 0.1 MB.
**TABLE S5**, PDF file, 0.2 MB.
**TABLE S6**, PDF file, 0.1 MB.
**TABLE S7**, PDF file, 0.1 MB.
**TABLE S8**, PDF file, 0.1 MB.

## ACKNOWLEDGMENTS

This work received funding from the National Natural Science Foundation of China (41806175) and National Key R&D Program of China (2018YFD0900106). As part of the EU Nano3Bio program, the authors are grateful for support from the European Union's Seventh Framework Program under grant agreement no. 613931. The Bowler laboratory is further supported by the French Government "Investissements d'Avenir" programs MEMO LIFE (ANR-10-LABX-54), Université de Recherche Paris Sciences et Lettres (Université PSL) (ANR-1253 11-IDEX-0001-02), and OCEANOMICS (ANR-11-BTBR-0008).

We are grateful for the technical contributions from Anne-Flore Deton-Cabanillas, Catherine Cantrel, and Yann Thomas in the Bowler laboratory. We thank TopEdit (www.topeditsci.com) for its linguistic assistance.

Z.S. and C.B. conceived and designed the research. Z.S. performed the experiments with the contribution of O.A. and V.B. in designing and conducting the activity assay. R.G.D. contributed to MMETSP data analysis and, together with L.T., provided technical support and helpful advice during all experiments. S.L. conducted statistical analysis and refined the presentation format of all charts. Z.S., F.R.J.V., and D.D. performed the bioinformatics analysis of *Tara* Oceans metagenomes and metatranscriptomes. Z.S. wrote the manuscript with input and editing from V.B., D.D., and C.B. All the authors revised the manuscript critically.

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
