## [Reviewer comments · mSystems]

Characterization of a marine diatom chitin synthase using a combination of meta-omics, genomics and heterologous expression approaches

Zhanru Shao, Osei Ampomah, Fabio Vieira, Richard Dorrell, Shaoxuan Li, Leila Tirichine, Vincent Bulone, Delin Duan, and Chris Bowler

Corresponding Author(s): Delin Duan, Institute of Oceanology Chinese Academy of Sciences

Review Timeline:

Submission Date:	December 6, 2022
Editorial Decision:	January 3, 2023
Revision Received:	January 5, 2023
Accepted:	January 16, 2023

Editor: Paul Wilmes

Reviewer(s): Disclosure of reviewer identity is with reference to reviewer comments included in decision letter(s). The following individuals involved in review of your submission have agreed to reveal their identity: Susheel Bhanu Busi (Reviewer #2)

Transaction Report:

DOI: <https://doi.org/10.1128/msystems.01131-22>

January 3, 2023

Dr. Delin Duan
Institute of Oceanology Chinese Academy of Sciences
Qingdao
China

Re: mSystems01131-22 (Characterization of a marine diatom chitin synthase using a combination of meta-omics, genomics and heterologous expression approaches)

Dear Dr. Delin Duan:

Thank you for submitting your manuscript to mSystems. We have completed our review and I am pleased to inform you that, in principle, we expect to accept it for publication in mSystems. However, acceptance will not be final until you have adequately addressed the reviewer comments.

Preparing Revision Guidelines

Sincerely,

Paul Wilmes

Editor, mSystems

Journals Department
Reviewer comments:

Reviewer #2 (Comments for the Author):

A job well done on making the necessary revisions. Indeed, this reviewer is pleased with the manner in which online tools and their respective parameters are described as well.

Reviewer #3 (Comments for the Author):

My concern is that only one line (line #27) i Phaeodactylum tricornutum has been properly described. In response to this the authors argue that a second line (line #13) do express TpCHS1 in Phaeodactylum tricornutum. qPCR and fusion protein localisation indicate some (weak) expression of the fusion protein. The problem still being that line #13 do not incorporate GlcNac (no enzymatic activity) have little abnormal phenotypes and appear to grow like wt (data not provided). The functional data from alga included are all from line #27 and should be validated. Line #13 express some GFP but it should also be shown to produce active TpCHS1 protein. The paradox being that accepting line#13 as a second line will make the conclusions flawed as the phenotype is deviating from that of line #27.

Re: mSystems01131-22 (Characterization of a marine diatom chitin synthase using a combination of meta-omics, genomics and heterologous expression approaches)

Dear Dr. Delin Duan:

Thank you for submitting your manuscript to mSystems. We have completed our review and I am pleased to inform you that, in principle, we expect to accept it for publication in mSystems. However, acceptance will not be final until you have adequately addressed the reviewer comments.

Dear Dr. Wilmes,

The authors are so glad to receive the decision letter after New Year holiday! We are grateful for all the constructive suggestions and comments from you and the reviewers to make this work more rigorous and qualified for publication in mSystems.

Please find the responses to the reviewers below and all the revisions in the marked-up manuscript.

Thank you again for all your efforts and time on our manuscript.

Happy New Year!

Kind regards,

Delin

Reviewer comments:

Reviewer #2 (Comments for the Author):

A job well done on making the necessary revisions. Indeed, this reviewer is pleased with the manner in which online tools and their respective parameters are described as well.

A: We sincerely thank the reviewer for his/her efforts on proposing very useful comments to improve our manuscript. To be clearer, we have also added a sentence in the DATA AVAILABILITY STATEMENT section to facilitate the readers' retrieval (Lines 606-608). ["The detailed parameters of the bioinformatics analysis for both online tools and local softwares have been uploaded to a Zenodo archive \(https://zenodo.org/record/7388969#.Y42vkKhByUI\) \(readme.txt\)."](https://zenodo.org/record/7388969#.Y42vkKhByUI)

Reviewer #3 (Comments for the Author):

My concern is that only one line (line #27) i *Phaeodactylum tricornutum* has been properly described. In response to this the authors argue that a second line (line #13) do express TpCHS1 in *Phaeodactylum tricornutum*. qPCR and fusion protein localisation indicate some (weak) expression of the fusion protein. The problem still being that line #13 do not incorporate GlcNac (no enzymatic activity) have little abnormal phenotypes and appear to grow like wt (data not provided). The functional data from alga included are all from line #27 and should be validated. Line #13 express some GFP but it should also be shown to produce active TpCHS1 protein. The paradox being that accepting line#13 as a second line will make the conclusions flawed as the phenotype is deviating from that of line #27.

A: We sincerely appreciate that the reviewer has proposed his/her comments to help us elucidate more details on our work, and approved the retention of OE in Pt. We understand that this part of work did not provide two strong positive transgenic cell lines as "perfect" as Sc OE strains. In order to let the readers have a clearer understanding of the ins and outs of the overexpression in Pt and make our conclusions more objective and convincing, we have revised the descriptions in

Abstract (Lines 50-54), in Results (Lines 263-270) and in Discussion (Lines 418-426).

* All the line numbers are from “Marked-up manuscript”, not “Clean manuscript”.

Lines 50-54:

Enhanced TpCHS1 expression could induce abnormal cell morphology and reduce growth rates in *P. tricornutum*, which might be ascribing to the inhibition of the G2/M phase. *S. cerevisiae* was proved to be a better system for expressing large amount of active TpCHS1, which effectively incorporated UDP-*N*-acetylglucosamine in radiometric *in vitro* assays.

Lines 263-270:

In situ observation of TpCHS1 protein is of great importance for its functional investigation. In this study, we verified that it was very challenging to do biolistic transformation on *Thalassiosira* cells. Considering the similar plastid evolution (secondary endosymbiosis) and structure (bounded by four membranes) between *Phaeodactylum* and *Thalassiosira*, together with the fact of *P. tricornutum* being a well-established lab model, we introduced a fusion of TpCHS1 with a C-terminal *eGFP* gene (TpCHS1-*eGFP*) into *P. tricornutum* by gene bombardment.

Lines 420-428:

S. cerevisiae was proved in our study to be an effective system for expressing large amount of active TpCHS1, which is a better model for CHS enzymatic activity detection compared with the overexpression in *P. tricornutum*. However, diatom cells could provide credible information on cell phenotype variation and potential location of protein. Choosing *S. cerevisiae* and *P. tricornutum* for heterologous transformation in this study has their own strengths. It is worth mentioning that this work is flawed ascribing to the lacking of a second strong transgenic *P. tricornutum* cell line as #27, and gene editing experiments will be necessary for further elucidation of TpCHS1 function.

January 16, 2023

Dr. Delin Duan
Institute of Oceanology Chinese Academy of Sciences
Qingdao
China

Re: mSystems01131-22R1 (Characterization of a marine diatom chitin synthase using a combination of meta-omics, genomics and heterologous expression approaches)

Dear Dr. Delin Duan:

Your manuscript has been accepted, and I am forwarding it to the ASM Journals Department for publication. For your reference, ASM Journals' address is given below. Before it can be scheduled for publication, your manuscript will be checked by the mSystems production staff to make sure that all elements meet the technical requirements for publication. They will contact you if anything needs to be revised before copyediting and production can begin. Otherwise, you will be notified when your proofs are ready to be viewed.

If you would like to submit a potential Featured Image, please email a file and a short legend to msystems@asmusa.org. Please note that we can only consider images that (i) the authors created or own and (ii) have not been previously published. By submitting, you agree that the image can be used under the same terms as the published article. File requirements: square dimensions (4" x 4"), 300 dpi resolution, RGB colorspace, TIF file format.

We recognize that the video files can become quite large, and so to avoid quality loss ASM suggests sending the video file via <https://www.wetransfer.com/>. When you have a final version of the video and the still ready to share, please send it to mSystems staff at msystems@asmusa.org.

Sincerely,

Paul Wilmes
Editor, mSystems

Journals Department
E-mail: mSystems@asmusa.org